# Application of an EMG-Rehabilitation Robot in Patients with Post-Coronavirus Fatigue Syndrome (COVID-19)—A Feasibility Study

**DOI:** 10.3390/ijerph191610398

**Published:** 2022-08-20

**Authors:** Ewa Zasadzka, Sławomir Tobis, Tomasz Trzmiel, Renata Marchewka, Dominika Kozak, Anna Roksela, Anna Pieczyńska, Katarzyna Hojan

**Affiliations:** 1Department of Occupational Therapy, Poznan University of Medical Sciences, 60-781 Poznan, Poland; 2Neurorehabilitation Ward, Greater Poland Provincial Hospital, 60-480 Poznan, Poland; 3Department of Physiotherapy, University of Health Science, 85-067 Bydgoszcz, Poland; 4Egzotech sp. z o.o., 44-100 Gliwice, Poland; 5Faculty of Automatic Control, Electronics and Computer Science, Silesian University of Technology, 44-100 Gliwice, Poland; 6Department of Rehabilitation, Greater Poland Cancer Centre, 61-866 Poznan, Poland

**Keywords:** SARS-CoV-2, exercises, physiotherapy, occupational therapy, hand grip strength

## Abstract

This pilot study aimed to assess the safety and feasibility of an EMG-driven rehabilitation robot in patients with Post-Viral Fatigue (PVF) syndrome after COVID-19. The participants were randomly assigned to two groups (IG—intervention group and CG—control group) in an inpatient neurological rehabilitation unit. Both groups were assessed on admission and after six weeks of rehabilitation. Rehabilitation was carried out six days a week for six weeks. The patients in the IG performed additional training using an EMG rehabilitation robot. Muscle fatigue was assessed using an EMG rehabilitation robot; secondary outcomes were changes in hand grip strength, Fatigue Assessment Scale, and functional assessment scales (Functional Independence Measure, Barthel Index). Both groups improved in terms of the majority of measured parameters comparing pre- and post-intervention results, except muscle fatigue. Muscle fatigue scores presented non-significant improvement in the IG and non-significant deterioration in the CG. Using an EMG rehabilitation robot in patients with PVF can be feasible and safe. To ascertain the effectiveness of such interventions, more studies are needed, particularly involving a larger sample and also assessing the participants’ cognitive performance.

## 1. Introduction

COVID-19 is a condition caused by a novel severe acute respiratory syndrome, which has been known for the last two years. A SARS-CoV-2 infection manifests itself through a wide spectrum of symptoms, from asymptomatic to life-threatening and possibly, fatal [1,2,3,4,5,6]. The course of a SARS-CoV-2 infection is highly individual. Most patients recover completely; however, approximately 20% experience long-term adverse effects, including fatigue (58%), headaches (44%), cognitive impairment (27%), excessive hair loss (25%), and dyspnoea (24%) [7]. Fatigue is one of the most common effects of COVID-19 [2] and can still exist days after the first symptom of this condition [8,9]. According to the World Health Organization International Classification of Diseases for Mortality and Morbidity Statistics, 11th Revision (IDC-11), Post-Viral Fatigue (PVF) is a neurological condition; the definition of PVF includes chronic fatigue syndrome and myalgic encephalomyelitis [10]. With similar symptoms characterizing both chronic fatigue syndrome and amyalgic encephalomyelitis, and with no clear knowledge about their pathogenesis and no consensus in terms of medical treatment, grouping these conditions under the definition of PVF helps to facilitate research in this matter [11]. The main symptoms of PVF are chronic fatigue, variable nonspecific myalgia, depression, and sleep disturbances [12].

PVF had previously been reported after the onset of other epidemics. In 2003, there was an outbreak of the SARS-CoV-1 virus, which caused the SARS epidemic. Tansey et al. [13] assessed the health outcomes of recovered patients after hospital discharge and found that more than half of their patients had experienced fatigue syndrome throughout their recovery: 64% reported fatigue at three months, 54% at six months, and 60% about twelve months after discharge [13]. Chronic fatigue was also often observed after other viral infections such as Epstein-Barr virus, cytomegalovirus, coxsackieviruses, and other coronaviruses [11].

Some COVID-19 patients subsequently develop PVF [14,15]. During the current pandemic, the most common symptom of post-viral sequelae was fatigue [7]. Kashif et al. [16] observed fatigue in recovered COVID-19 patients three months after hospital discharge. PVF was also present in patients with a mild course of the disease [16,17]. It is hypothesized that PVF in Post-COVID patients results from building up cytokines, which affects the central nervous system’s lymphatic drainage—the “glymphatic system” [18].

The symptoms that patients reported after COVID-19 were not limited to those typical of PVF. The most common Post-COVID complications were those related to patients’ physical and cognitive functioning. Very often, those symptoms were dominated by neuromuscular and immune exhaustion such as a lack of strength (as a polyneuropathy), disorders of neurological functions (such as cognitive disorders), aches and pains, problems with sleeping but also those with moving and receiving sensory stimuli, disorders of the immune system, and functions of the urogenital and digestive systems [19,20,21].

To cover the symptoms of patients after recovery from COVID-19, the synonymous terms “Long-COVID” and “Post-COVID Syndrome” are used. There is no clear, unanimous definition of Long-COVID [22]. According to Raveendran et al., “Long-COVID” is the presence of various symptoms, even weeks or months after the SARS-CoV-2 infection, and the viral status is not related to those symptoms [23]. These authors proposed the term “Long-COVID” for both post-acute COVID-19 patients (from 3 weeks up to 3 months after infection) and chronic ones (beyond 3 months after infection). The National Institute for Health and Care Excellence (NICE) guidelines proposed the definitions of “Acute COVID” (when the signs and symptoms of COVID-19 are present for up to 4 weeks), “Ongoing COVID” (when the signs and symptoms of COVID-19 are present from 4 weeks up to 12 weeks), and “Post-COVID Syndrome” (when the symptoms last more than 12 weeks and cannot be explained by other underlying diseases). Post-COVID Syndrome, according to these guidelines, can also be identified before the threshold of 12 weeks when a possibility of other underlying diseases is excluded. The NICE guidelines also introduced the definition of “Long-COVID,” which is a term that includes both Ongoing COVID and Post-COVID Syndrome.

Due to a lack of unified definitions, PVF and Long-COVID pose a challenge to modern medicine. There is a growing need for research and the development of efficient therapeutic options for patients. Post-COVID rehabilitation is usually comprised of light aerobic and breathing exercises. Still, patients’ needs may vary from case to case. Notably, patients with myalgic encephalomyelitis or chronic fatigue syndrome may present adverse responses to exercise. These patients may thus need a different approach [22]. The present pandemic showed that, especially in the wake of a shortage of medical staff, robotic devices could constitute a good option for the support and rehabilitation of COVID-19 patients [24,25]. Therefore, the aim of this paper was to assess the safety and feasibility of the use of an EMG-Rehabilitation Robot in exercises performed by Post-COVID patients with PVF.

## 2. Materials and Methods

### 2.1. Study Design

A pilot prospective clinical study was conducted in the Neuro-rehabilitation Ward of the Provincial Hospital in Poznan, Poland, between January and November 2021.

### 2.2. Study Group

In our study, we included 30 adults admitted to the inpatient neurological rehabilitation unit. The sample size was decided based on previous pilot studies concerning biofeedback and/or robotic interventions [26,27,28]. Each participant was recruited with informed consent to participate in the study.

### 2.3. Study Criteria

We included participants, all of whom were transferred directly from Intensive Care Units (ICU) due to PVF after experiencing a severe course of COVID-19, according to the criteria presented by Carod-Artal [29]. The exclusion criteria comprised additional diagnosis of another active infection process, neoplastic, rheumatic, metabolic, endocrine, autoimmune, and cardiovascular diseases, as well as conditions for which constant fatigue is typical: multiple sclerosis, systemic lupus or Hashimoto’s disease, common hypothyroidism, Lyme disease, AIDS, diabetes, or myasthenia gravis, psychiatric disorders except for unipolar depressed moods.

The study participants were randomly assigned to two groups (IG—intervention group and CG—control group), thus constituting a 1:1 randomized study (allocation based on the selection of the hospital computer). The total stay of each patient in the Rehabilitation Unit was six weeks. Both groups were assessed twice: on admission and after six weeks.

### 2.4. Outcome Measure

The main outcome measure was muscle fatigue during the two types of rehabilitation treatment. Additionally, the functional changes after rehabilitation were analyzed after two types of neuro-rehabilitation.

### 2.5. Measurement

Sociodemographic data were collected from all subjects; additionally, the following functional assessment tools were used: Functional Independence Measure (FIM), Barthel Index (BI), handgrip strength (HGS), Fatigue Assessment Scale (FAS), and muscle fatigue assessment model using the LUNA Rehabilitation Robot (EGZOTech, Poland, registration number/TNP/MDD 0373/4038/2021).

The Functional Independence Measure (FIM) is used to assess the patient’s level of disability and a change in their status in response to rehabilitation or medical intervention. FIM is an 18-item instrument that comprises measures of independence for self-care, including transfers, locomotion, communication, sphincter control, and cognition [30]. Each item is scored on a 7-point ordinal scale (ranging from 1 to 7). The higher the score, the more independent the patient is in performing the task associated with that item. In addition, the score reflects the level of assistance an individual needs—from total independence to total assistance [31].

The Barthel Index is an ordinal scale used to measure performance in activities of daily living (ADL). BI measures the degree of assistance required by an individual on ten items of mobility and self-care activities: feeding, personal toileting, bathing, dressing and undressing, moving on and off a toilet, controlling the bladder, controlling bowel, moving from wheelchair to bed and returning, walking on a level surface and ascending and descending stairs. The Index is a three-item ordinal rating scale, and each item is rated in terms of whether the patient can perform the task independently-2 points, with some assistance-1 point, or is dependent on help 0 points. The maximum final score is 20 (assistance in ADL is not needed) [32].

Handgrip strength was measured using a JAMAR Hand Dynamometer (Sammons Preston Rolyan, USA). The measurement was performed in a sitting position in an armless chair, with the forearm in a 90-degree flexed position. The average of three measurements (in kilograms) was recorded [33].

Fatigue Assessment Scale (FAS) is a 10-item fatigue questionnaire to assess fatigue in the general population. Five questions reflect physical fatigue, and another five (questions 3 and 6–9)—mental fatigue. The total score ranges from 10 to 50. A total FAS score of <22 indicates no fatigue, and a score of ≥22 indicates the presence of fatigue [34].

Muscle fatigue data were also retrieved from the EMG-rehabilitation robot regarding the work of the muscles of the two-headed arm by means of the robot’s muscle fatigue assessment model (isometric contraction tension). The test of EMG-based fatigue was performed on the Biceps Brachii muscle by surface electrodes. The electrodes were placed on the line between the medial acromion and the fossa cubit at 1/3 from the fossa cubit according to SENIAM [35]. The sampling frequency of the EMG signal was 1000 Hz. The resulting data were filtered using a bandpass filter and an optional notch filter (50 Hz) to prevent power line interferences. The upper limb extension was set and locked at 90 degrees of elbow flexion. The protocol of the test included 30 s of relaxation, then 30 s of contraction, and 30 s of relaxation. If the patient was not able to perform 30 s of contraction, the algorithm calculated the time when the muscle was in contraction based on the EMG signal only. Based on the raw data exported from the robot, frequency analysis was performed using external Python-based software, yielding isokinetic force and EMG (real-time activity) surface signal analysis.

### 2.6. Rehabilitation

Neurological rehabilitation in both study groups was carried out six days a week for six weeks using neuromuscular re-education techniques (such as PNF—Proprioceptive Neuromuscular Facilitation and Bobath therapy), exercises for movement coordination and balance, with combined progressive endurance training. After the participants achieved the ability to maintain a standing position and conduct active and resistance exercises, progressive endurance training was performed from 35 percent of max HR (heart rate) to 70 percent of max HR (cycling and walking) according to the equation max HR = 220 − age 30 min per day. PNF or Bobath therapy was conducted 45 min per day. The patients in the IG had this training for 75 min per day, and the CG had this form of therapy for 120 min a day.

Each patient (in both IG and CG) worked individually with a psychologist and speech therapist five times a week for 30 min. Additionally, occupational therapy was performed daily for a minimum of 30 min.

The patients in the IG, according to the intervention model of rehabilitation, performed additional training using an EMG rehabilitation robot—the EMG-rehabilitation robot (two sessions of a total of 45 min a day for six days a week). The EMG-rehabilitation robot enables the isokinetic-isotonic training of selected muscle groups using reactive electromyography (EMG-triggered), providing assisting robot movement, which is activated by the patient’s muscle bioelectric activity signal measured in real-time with surface electrodes. During exercises with the EMG robot, the participants conducted flexion and extension of the elbow and flexion, extension, and abduction in the glenohumeral joint. The muscles of the upper limb (such as biceps brachii, triceps brachii, and deltoid muscle) were used to trigger the movement. Figure 1 presents the setup of the robotic device during one of the exercises. All of the EMG procedures were carried out in accordance with the SENIAM guidelines [35].

Luna EMG is a rehabilitation robot specifically designed to support the physiotherapy of neurological patients with muscle weakness. It is an all-in-one platform for complex personalized therapy for patients suffering from neurological conditions. It tackles clinical issues such as muscle weakness, mobility disorders, gait problems, and range of motion restrictions, specifically by automating the process of personalized, motivating physiotherapy based on electromyography combined with force and position sensing. The robot was attached to the patient through extensions–interchangeable mechanical parts that are connected to the patient by straps or by the grip. The movements are controlled by a Windows application from a therapist panel, which provides a User Interface, patient management, reporting module, and Internet connectivity for the purpose of remote diagnostics and oversight.

The device performs isokinetic, isotonic, and isometric exercises. The “EMG-triggered robotic movement” technology works actively with patients, even if no movement is observable. The EMG-rehabilitation robot detects the EMG activity of the muscle and provides appropriate assistance during the movement. If no movement or muscle activity is present, the device provides passive assistance.

The EMG-rehabilitation robot follows a “hands-off” methodology to improve the independence of the patient, even when their mobility and muscle activity are very limited. In the wake of the COVID-19 pandemic, this approach seems to be beneficial since it limits direct contact between the patient and the therapist while maintaining proper rehabilitation measures.

## 3. Statistical Analysis

Statistical analysis was performed with the Statistica 13.3 software (TIBCO Software, Warszawa, Poland). The normality of the distribution was checked with the Shapiro-Wilk test. In the absence of a normal distribution and for ordinal variables, the Mann–Whitney test was used to compare the results in two groups. The Wilcoxon signed-rank test was used for intragroup comparisons. Data are presented as medians and range. The χ^2^ test was used to compare two groups of nominal variables. *p* < 0.05 was considered statistically significant.

## 4. Results

A total of 30 participants (15 in IG and 15 in CG) were included (Figure 2 presents the flow diagram of the study). Data from two participants (one from each group) regarding muscle fatigue were not available; therefore, the analysis of this particular outcome comprised data from 28 participants.

No statistically significant differences were found between group characteristics (concerning age, sex, weight, height, BMI, time in the ICU, and time intubated) at baseline (Table 1).

Both the IG and CG improved in terms of the majority of measured parameters comparing the pre- and post-intervention results, except for muscle fatigue measured by EMG. Muscle fatigue scores presented non-significant improvement in IG and non-significant deterioration in CG. The results of the pre-post comparison are presented in Table 2.

The comparison of mean changes of measured parameters did not reveal any statistically significant differences between the study groups. The results are presented in Table 3.

The anonymized EMG charts from pre- and post-intervention measurements of all of the participants are included in the Appendix A. No adverse effects of the exercises on EMG—rehabilitation robot (such as musculoskeletal pain, numbness, or deterioration of physical function) were reported by the participants. All of the participants were able to perform assigned EMG—rehabilitation robot exercises daily for a full 45 min.

## 5. Discussion

In this pilot study, the feasibility of the concept of using the EMG—rehabilitation robot in exercises for patients with PVF syndrome was verified positively. The study participants reported no adverse outcomes, and no dropouts from the study were observed. The lack of data from two participants resulted from transferring them to another facility. It must be noted that not only no adverse outcomes were reported but also improvement in both groups was observed. Based on that, conducting further research on this topic following the methodology presented in this study can be safe for patients and is not likely to negatively impact their outcomes of rehabilitation.

The use of robots in rehabilitation started on a larger scale in 1990 [36], and this branch of medicine and technology is still developing. Robotic devices have many advantages, among them the ability to maintain precise continuous movement and the potential for actively engaging users [36]. These features make them valuable in the rehabilitation process and deliver the base for efforts to implement the robots in a variety of conditions, with stroke leading among them. Villafañe et al. [37] conducted a study of the efficacy of robot-assisted rehabilitation in stroke patients with hand paralysis. The experimental group received passive mobilization by a robotic device in addition to their regular physical therapy and occupational therapy, while the control group performed more physical and occupational therapy instead. The authors reported greater improvement in the experimental group and concluded that robotic rehabilitation could be beneficial in regard to pain and spasticity. Bustamante-Valles et al. [38] performed a study comparing traditional therapy after stroke and robot-assisted circuit training, concluding that robot-assisted therapy can be as efficient as traditional occupational or physical therapy. The authors also showed that the cost of 2 h of conventional therapy to the national healthcare system in Mexico was 19.21 USD, while 2 h of robot-assisted therapy cost only 6.99 USD. The authors’ conclusion was that involving robots in the rehabilitation process is not only substantively effective but also cost-effective compared to conventional rehabilitation or occupational therapy. In 2017, a systematic review, including ten trials involving 502 participants, showed that robot-assisted gait training improved gait parameters, participants’ mobility, and independence significantly greater than conventional gait training in patients with incomplete spinal cord injury [39].

As Yong [22] stated, among all of the available therapeutic options, only rehabilitation seems to be effective in the treatment of symptoms of Long-COVID. Many studies focus on Post-COVID symptoms or the need for rehabilitation [37,38,39,40] or propose guidelines and rehabilitation programs [40,41,42], but there is a distinct lack of studies assessing rehabilitation effectiveness in patients with PVF [12,43]. Udina et al. [44] conducted a study on the effect of Post-COVID rehabilitation on COVID-19 survivors’ independence and physical performance. In their study, during the rehabilitation, the patients were obliged to perform 30 min of exercises daily (involving resistance, endurance, balance, and breathing exercises). The authors demonstrated that exercises improved participants’ independence and physical performance. Spielmanns et al. [43] also assessed the effect of Post-COVID rehabilitation on patients’ physical performance and independence. The authors reported improvement in regard to FIM scores after rehabilitation in both the Post-COVID group (from 100 (±15.1) in pre- to 111 (±15.0) in post-rehabilitation measurements) and the control group, which was comprised of patients with lung diseases–from 99.7 (±9.72) to 107 (±10.7). These results are consistent with ours regarding the functional independence indices. Nopp et al. [45] measured fatigue by FAS, similarly to the present study. The authors observed improvement in terms of FAS scores and 6-min walk test after six weeks of individualized pulmonary rehabilitation. Their protocol of rehabilitation contained endurance, strength, and respiratory muscle training alongside psychologic, dietary, and nutritional interventions. Daynes et al. [46] investigated post-COVID-19 rehabilitation efficiency in improving fatigue symptoms, breathlessness, exercise capacity, and cognition. Thirty individuals attended twelve rehabilitation sessions each (two per week). Each session consisted of physical rehabilitation (endurance, aerobic, and strengthening exercises) and patients’ education, which covered various topics linked with pulmonary functioning, such as fatigue, cough, breathlessness, smell, and other conditions related to COVID-19 symptoms (i.e., loss of taste, cognition problems, and anxiety). The participants showed improvement in regard to fatigue. Muscle fatigue was measured in none of the abovementioned studies [43,44,45,46]. To the best of our knowledge, the present study is the first one tackling the topic of using a robotic device in the PVF syndrome. Therefore, comparing our results in this particular domain to the literature is difficult.

Our study demonstrated that implementing an EMG robot-rehabilitation protocol in PVF patients is feasible and safe. Similar results were obtained by different authors in studies using comparable devices. Kim et al. [47] conducted a study on 12 stroke survivors who were exercising on an electromyography-triggered hand robot. The authors observed an improvement concerning hand function indices with no serious adverse effects. Stroke patients were also the target of a feasibility study conducted by Singer et al. [48]. The authors observed that EMG- triggered electrostimulation was feasible, safe, and efficient in improving stroke patients’ hand function.

It should be noted that our study has some limitations. First of all, the lack of significant difference in terms of improvement between the IG and CG could result from the too-small sample size. As the present study aimed to assess the feasibility and safety of the presented protocol, further research involving larger samples is needed to establish the actual influence of exercises on a rehabilitation robot on muscle fatigue and fatigue-related outcomes. A further limitation is that PVF is often accompanied by cognitive disorders, and exercises on the EMG—rehabilitation robot required notable focus and psychological effort to perform the exercises and measurements correctly. Further studies are needed to analyze if exercising on an EMG—rehabilitation robot affects cognitive abilities. The third limitation results from the fact that this study is, to the best of our knowledge, the first one assessing an EMG-driven device in the therapy of PVF. While it can be perceived as a strength of the study, it must be pointed out that there exists no literature with which to relate our results. While most of the studies assessing the effectiveness of such devices target stroke patients, due to the different nature and causes of PVF and stroke-related impairments, the effectiveness of such devices can differ between those groups. The different exercise regimens in the studied groups can also be viewed as a limitation.

## 6. Conclusions

According to the results obtained in this feasibility study, using an EMG-robot in the rehabilitation of patients with PVF can be feasible and safe. To ascertain the effectiveness of such interventions in this group of patients, more studies are needed, particularly involving a larger sample and also assessing the participants’ cognitive performance.

## Figures and Tables

**Figure 1 ijerph-19-10398-f001:**
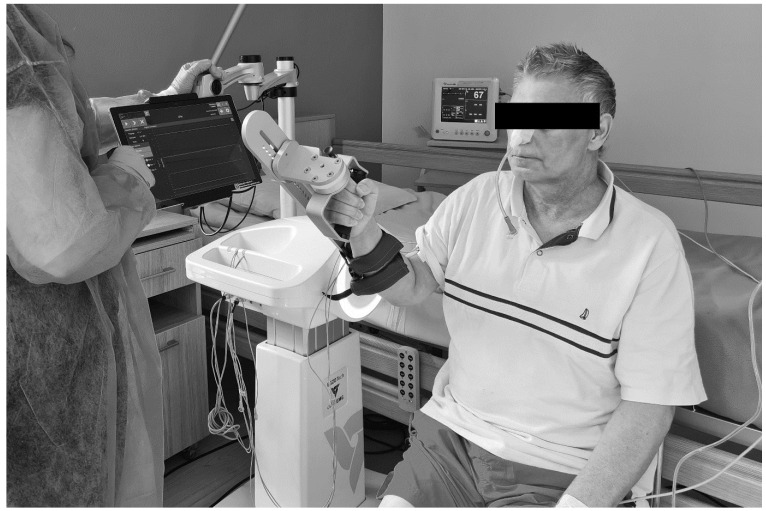
An example of the EMG robot setup during the exercises (source: authors’ own).

**Figure 2 ijerph-19-10398-f002:**
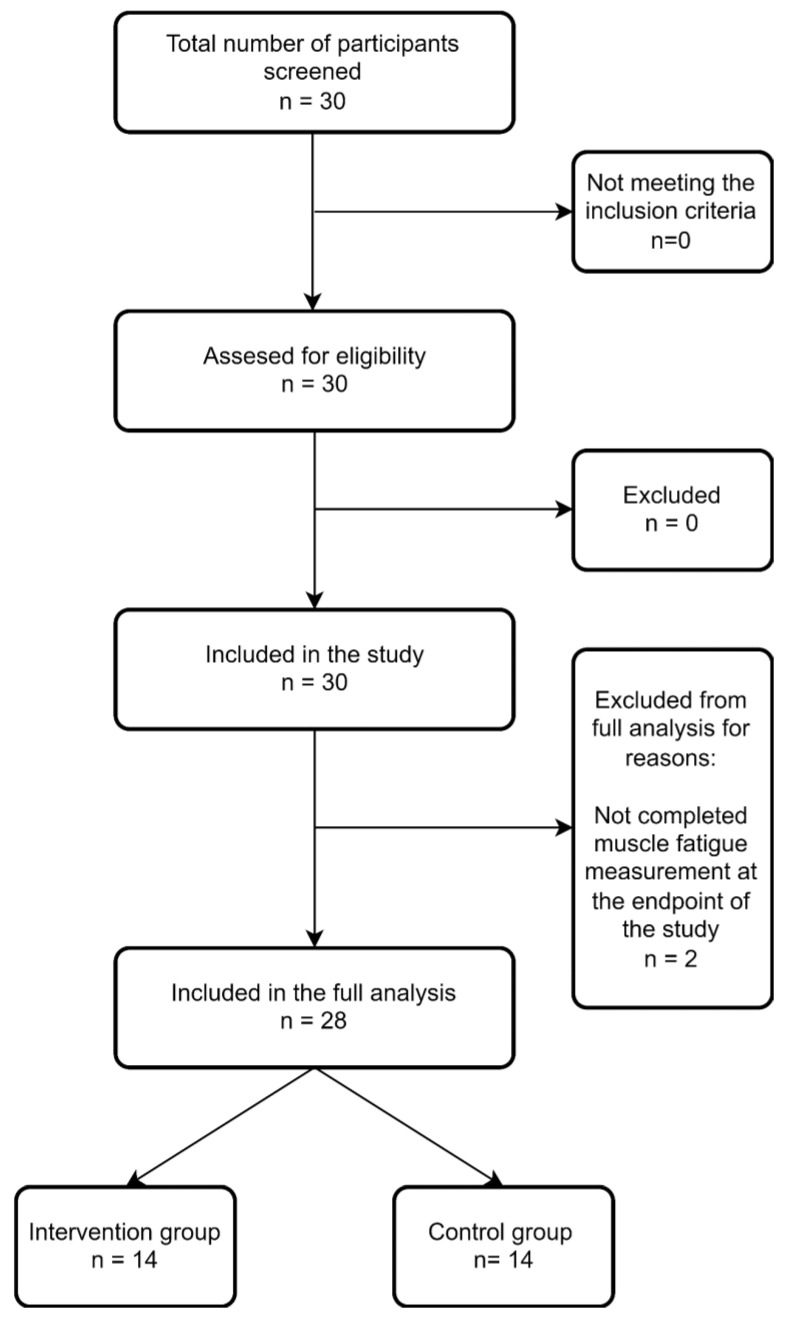
Study participants flowchart.

**Table 1 ijerph-19-10398-t001:** Baseline group characteristics.

	Intervention Group	Control Group	*p*-Value
Sex (%) *			0.439
Female	40	26.67	
Male	60	13.33	
Age (years) +	69 (43–81)	66 (39–75)	0.372
Level of education % *			0.659
primary	6.67	0	
vocational	26.67	40	
secondary	46.67	46.67	
high	20	13.33	
Height (cm) +	171 (150–188)	173 (154–192)	0.442
Weight (kg) +	77 (50–103)	80 (49–97)	0.561
BMI +	26.33 (20.81–32.32)	25.25 (19.14–36.07)	0.130
Time in the ICU (days)	25 (20–41)	24 (20–44)	0.917
Time intubated (days)	23(15–35)	21(15–33)	0.604

*—data presented as percentage of overall group population; +—data presented data presented as median and range; ICU—intensive care unit.

**Table 2 ijerph-19-10398-t002:** Comparison of the pre- and post-intervention results.

Outcome Measure (Median and Range)	Intervention Group			Control Group		
	Pre-Intervention	Post-Intervention	*p*	Pre-Intervention	Post-Intervention	*p*
FIM	85 (8–120)	117 (78–136)	0.001	89 (32–120)	117 (5–126)	0.005
HGS	18 (0–35)	20 (1–37)	0.001	20 (10–39)	22 (14–40)	0.007
BI	11 (2–14)	18 (15–20)	0.001	12 (3–14)	19 (2–20)	0.001
FAS	25 (15–42)	23 (7–38)	0.001	26 (14–42)	26 (13–48)	0.041
Fatigue (EMG)	−5.95 (−29.2–5.4)	−6.8 (−17.6–20.9)	0.778	−2.2 (−20.1–43.6)	−1.05 (−22.2–10.2)	0.975

FIM—Functional Independence Measure, HGS—Handgrip strength, BI—Barthel Index, FAS—Fatigue Assessment Scale, Fatigue (EMG)—muscle fatigue calculated from EMG measurement data, expressed as a percentage of the slope of the frequency curve.

**Table 3 ijerph-19-10398-t003:** Comparison of mean pre-post changes of outcomes between groups.

Outcome Measure (Median and Range)	Intervention Group	Control Group	*p*
FIM	26 (16–113)	23 (−27–54)	0.137
HGS	3 (0–10)	4 (−9–10)	0.367
BI	8 (4–14)	6 (−3–11)	0.233
FAS	−2 (−11–0)	−2 (−7–7)	0.412
Fatigue (EMG)	0 (−14.9–34.7)	2.8 (−55.4–11.3)	0.909

FIM—Functional Independence Measure, HGS—Handgrip strength, BI—Barthel Index, FAS—Fatigue Assessment Scale, Fatigue (EMG)—muscle fatigue calculated from EMG measurement data, expressed as a percentage of the slope of the frequency curve.

## Data Availability

The authors confirm that the data supporting the findings of this study are available within the article, its’ Appendix A and are available from the corresponding author, [T.T.], upon reasonable request.

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
