# Peer review of "Application of an EMG-Rehabilitation Robot in Patients with Post-Coronavirus Fatigue Syndrome (COVID-19)—A Feasibility Study"

_ijerph, 2022, doi:10.3390/ijerph191610398_

Round 1
Reviewer 1 Report
This study presents an innovative approach to Post-Viral Fatigue syndrome after COVID-19. It assess the safety and feasibility of an EMG-driven rehabilitation robot these patients which is a new approach. These are my comments:
My general advice is to put references before period, in the sentence, not after the period.
Introduction:
When you mention previous PVF occurence following the outbreak of SARS-CoV virus in 2003 I believe more correct term for that virus is SARS-CoV-1 virus.
Methods:
How did you estimate your sample size? On what ground?
When you describe your study population you state that they were transferred to the rehabilitation facility directly from the intensive care unit. It is unclear whether they only had PVF or they were in general in weakened state due to being treated in intensive care for some time. Most patients feel fatigue after being treated in intensive care, it does not have to be PVF. From the description of therapies used (speech therapy etc.) it seems it was not only the case of PVF.
Last section of Measurement chapter - there is space missing between the words robot and regarding. Same thing happens in the last section of Discussion.
Please give more detailed information regarding EMG data measured by the robot - which muscles, a what anatomic points did you measure the signal, how exactly did you record the signal, at what frequency, how did you process the EMG signal. For the CG the robot was used only as an assessment method? Is it valid that way?
What kind of progressive endurance training did you use? Also, give some data regarding the therapies - how many minutes of Bobath, PNF, endurance training per day? You mention just total time of therapeutic exercise/neurorehabilitation techniques.
Your IG had 75 min of therapeutic exercise per day and your CG only 20 minutes. This is a big difference! The difference between the groups could be resulted just from this difference, and not by additional training using EMG rehabilitation robot. Description of specific exercises and/or movement patterns during robot therapy is not very clear. What exercises did the patients perform?
Did you check normality of the data? I see that you used non-parametric tests, but your descriptive statistics is presented as mean and SD. Why not median and IQR if the data is non normal?
Results
Baseline group characteristics - you did not mention low long were the patients in the intensive care unit? How many of them intubated? This could make a big difference if there is a difference between the groups at the start of the trial.
Based on the data provided I am not quite sure that your results are derived just from the robot intervention. Your groups could have been different at the end of the trial due to different therapeutic exercise regimen in CG and IG, and also due to their different baseline health status (different length of stay in ICU). EMG robot intervention is only a third option...
Discussion
I am aware that there is a lack of similar studies to compare the results. However, you discuss only three studies regarding Post-Covid rehabilitation, and there are many studies. Your Discussion should be stronger.
Author Response
My general advice is to put references before period, in the sentence, not after the period.
Answer: Changed according to reviewer sugestion
Introduction:
When you mention previous PVF occurence following the outbreak of SARS-CoV virus in 2003 I believe more correct term for that virus is SARS-CoV-1 virus.
Answer: Changed according to reviewer sugestion
Methods:
How did you estimate your sample size? On what ground?
Answer:
We decided that 30 participants will be suficient to detect potencial issues during the pilot study, based on previous pilot studies concerning biofeedback and/or robotic intervention. Information has been added in the manuscript.
- Tamburella, F.; Moreno, J.C.; Herrera Valenzuela, D.S.; Pisotta, I.; Iosa, M.; Cincotti, F.; Mattia, D.; Pons, J.L.; Molinari, M. Influences of the biofeedback content on robotic post-stroke gait rehabilitation: electromyographic vs joint torque biofeedback. J. Neuroeng. Rehabil. 2019, 16, 95, doi:10.1186/s12984-019-0558-0.
- Middaugh, S.; Thomas, K.J.; Smith, A.R.; McFall, T.L.; Klingmueller, J. EMG Biofeedback and Exercise for Treatment of Cervical and Shoulder Pain in Individuals with a Spinal Cord Injury: A Pilot Study. Top. Spinal Cord Inj. Rehabil. 2013, 19, 311–323, doi:10.1310/sci1904-311.
- Calabrò, R.S.; Filoni, S.; Billeri, L.; Balletta, T.; Cannavò, A.; Militi, A.; Milardi, D.; Pignolo, L.; Naro, A. Robotic Rehabilitation in Spinal Cord Injury: A Pilot Study on End-Effectors and Neurophysiological Outcomes. Ann. Biomed. Eng. 2021, 49, 732–745, doi:10.1007/s10439-020-02611-z.
When you describe your study population you state that they were transferred to the rehabilitation facility directly from the intensive care unit. It is unclear whether they only had PVF or they were in general in weakened state due to being treated in intensive care for some time. Most patients feel fatigue after being treated in intensive care, it does not have to be PVF. From the description of therapies used (speech therapy etc.) it seems it was not only the case of PVF.
Answer: The study patients were transferred to the neurological rehabilitation department on a referral from an anesthesiologist and neurologist as PVF - post-covid fatigue syndrome.
As there is a speech therapist in our ward, and most of the patients had trouble swallowing/dysphasia, they were also looked after by a speech therapist.
Last section of Measurement chapter - there is space missing between the words robot and regarding. Same thing happens in the last section of Discussion.
Answer: Corrected as sugested by the Reviewer
Please give more detailed information regarding EMG data measured by the robot - which muscles, a what anatomic points did you measure the signal, how exactly did you record the signal, at what frequency, how did you process the EMG signal. For the CG the robot was used only as an assessment method? Is it valid that way?
Answer: Detailed information regarding EMG data measured has been added as sugested by the reviewer.
The EMG signal was measured on Biceps brachii by surface electrodes. Electrodes were placed on the line between the medial acromion and the fossa cubit at 1/3 from the fossa cubit according to SENIAM. Sampling frequency of the EMG signal was 1000 Hz. Data is filtered using a bandpass filter and an optional notch filter to power line interference (50Hz).
What kind of progressive endurance training did you use? Also, give some data regarding the therapies - how many minutes of Bobath, PNF, endurance training per day? You mention just total time of therapeutic exercise/neurorehabilitation techniques.
Answer: Detailed information has been added.
After participants achieved ability to maintain standing position and conduct active and resistance exercises, progressive endurance training was performed from 35max HR to 70 max HR ( cycling, and walkig) according to HR=220- age 30 min /day. PNF or Bobath therapy was conducted 45 min per day.
Your IG had 75 min of therapeutic exercise per day and your CG only 20 minutes. This is a big difference! The difference between the groups could be resulted just from this difference, and not by additional training using EMG rehabilitation robot. Description of specific exercises and/or movement patterns during robot therapy is not very clear. What exercises did the patients perform?
Answer: We apologize this error occured during during translation and preparing to submission, in manuscript in our native language was stated 120 not 20 minutes. We apologise for this oversight and we are gratefull that Reviewer spoted this issue.It has been corrected in the manuscript.
Did you check normality of the data? I see that you used non-parametric tests, but your descriptive statistics is presented as mean and SD. Why not median and IQR if the data is non normal?
Answer: Changed as sugested by the reviewer
Results
Baseline group characteristics - you did not mention low long were the patients in the intensive care unit? How many of them intubated? This could make a big difference if there is a difference between the groups at the start of the trial.
Answer: Information regarding lenght of intubation and stay in ICU has been added. No statistical differences in terms of those data were observed.
Based on the data provided I am not quite sure that your results are derived just from the robot intervention. Your groups could have been different at the end of the trial due to different therapeutic exercise regimen in CG and IG, and also due to their different baseline health status (different length of stay in ICU). EMG robot intervention is only a third option...
Answer: Participants did not differ in term of length of stay in ICU and intubation and total time of rehabilitation. They also did not differ in term of improvement between groups.
Discussion
I am aware that there is a lack of similar studies to compare the results. However, you discuss only three studies regarding Post-Covid rehabilitation, and there are many studies. Your Discussion should be stronger.
Answer: As sugested by the reviewer Discussion has been expanded
Reviewer 2 Report
Thank you to the authors. All sections especially method section of the article was written in detail.. I read carefully.. Almost every detail was mentioned in methods section.. I accepted the article in current form..
Author Response
Suggestions for Authors
Thank you to the authors. All sections especially method section of the article was written in detail.. I read carefully.. Almost every detail was mentioned in methods section.. I accepted the article in current form.
Answer: Thank you for your comment.
Reviewer 3 Report
Application of an EMG - Rehabilitation Robot in Patients With Post-Coronavirus Fatigue Syndrome (COVID-19) – a feasibility study investigates the feasibility of a robotic-based approach for treating patients affected with Post-Viral Fatigue (PVF) syndrome after COVID-19. The paper is within the scope of the journal and in general clear to read.
I have some comments to improve the quality of the paper.
Fontsize is not always uniform (e.g.: abstract).
Abstract should be improved, for example with using more punctuation. Trial number is usually not reported in the abstract but in the methods.
The rationale for robotic intervention should probably be supported more strongly. Was robot-therapy effective with similar fatigue syndromes previously? How EMG based robots or analyses helped in robotic treatments?
Some critical elements are missing: which muscles were recorded or used for triggering? What control logics was followed to trigger movement? What tasks were performed? A photo of the robotic setup while in use would be useful.
Figure 1 must be done in vectorial format, it is very hard to read and it is of poor quality.
Is it correct that a group of participants had “additional intervention”? it seems that the control group had less therapy… probably it would be better to had different interventions (robot vs standard) rather than more, otherwise results are not really comparable. Please comment on this.
Are FAS test p-value so low, despite high standard deviation? Can you please check?
Please, explain how you computed EMG based fatigue (condition, duration, and at least basic concepts for the algorithm).
Author Response
Application of an EMG - Rehabilitation Robot in Patients With Post-Coronavirus Fatigue Syndrome (COVID-19) – a feasibility study investigates the feasibility of a robotic-based approach for treating patients affected with Post-Viral Fatigue (PVF) syndrome after COVID-19. The paper is within the scope of the journal and in general clear to read.
I have some comments to improve the quality of the paper.
Fontsize is not always uniform (e.g.: abstract).
Answer: Corrected as suggested.
Abstract should be improved, for example with using more punctuation. Trial number is usually not reported in the abstract but in the methods.
Answer: Abstract has been changed
The rationale for robotic intervention should probably be supported more strongly. Was robot-therapy effective with similar fatigue syndromes previously?
Answer:There are no previous studies in that matter.
How EMG based robots or analyses helped in robotic treatments?
Some critical elements are missing: which muscles were recorded or used for triggering? What control logics was followed to trigger movement? What tasks were performed? A photo of the robotic setup while in use would be useful.
Answer: Detailed information of EMG measurement. Also information regarding exercises setup and photo of the device while in use was added according to suggestion.
Figure 1 must be done in vectorial format, it is very hard to read and it is of poor quality.
Answer: Figure 1 quality has been improved as suggested by the reviewer.
Is it correct that a group of participants had “additional intervention”? it seems that the control group had less therapy… probably it would be better to had different interventions (robot vs standard) rather than more, otherwise results are not really comparable. Please comment on this.
Answer: We apologize this error occured during translation and preparation to submission, in manuscript in our native language was stated 120 not 20 minutes of rehabiliation in CG. It has been corrected.
Are FAS test p-value so low, despite high standard deviation? Can you please check?
Please, explain how you computed EMG based fatigue (condition, duration, and at least basic concepts for the algorithm).
Answer: FAS was checked by our statistician. Detailed information regarding EMG based fatigue has been added.
The test of EMG based fatigue was performed on Biceps Brachii muslce. The upper limb extension was set and blocked on 90 degree of elbow flexion. The test of EMG-based fatigue was performed on the Biceps Brachii muscle. The position of upper limb extension was set and blocked on 90 degrees of elbow flexion. The protocol of the test has 30 seconds of relaxation, the next 30s of contraction, and 30 seconds of relaxation. If the patient is not able to performed 30 second of contraction, the algorithm takes into calculation only time when the muscle was in contraction based on the EMG signal.

Round 2
Reviewer 1 Report
I am happy with the improvements of the article. However, different therapeutic exercise regimen between the groups should be stated as a limitation of the study.
Author Response
we agree that different exercise regimens can be treated as a limitation of the study. A corresponding sentence has been added to the manuscript text.
Reviewer 3 Report
The authors have addressed my main concerns. I still think that the paper might benefit from a careful re-read and little improvements (e.g.: Fig. 1 is still quite low-quality, at least in the draft I see).
Author Response
We believe that the technical quality of Figure 1 is more than sufficient. If you still need a bigger image file, please let me know.